# Ophthalmic Evaluation of Diagnosed Cases of Eye Cystinosis: A Tertiary Care Center’s Experience

**DOI:** 10.3390/diagnostics10110911

**Published:** 2020-11-07

**Authors:** Malgorzata Kowalczyk, Mario Damiano Toro, Robert Rejdak, Wojciech Załuska, Caterina Gagliano, Przemyslaw Sikora

**Affiliations:** 1Department of General Ophthalmology and Pediatric Ophthalmology Service, Medical University of Lublin, 20079 Lublin, Poland; makowal@op.pl (M.K.); robertrejdak@yahoo.com (R.R.); 2Faculty of Medicine, Collegium Medicum Cardinal Stefan Wyszyński University, 01815 Warsaw, Poland; 3Department of Nephrology, Medical University of Lublin, 20954 Lublin, Poland; wojciech.zaluska@umlub.pl; 4Ophthalmology Clinic, San Marco Hospital, University of Catania, 95123 Catania, Italy; caterina_gagliano@hotmail.com; 5Department of Pediatric Nephrology, Medical University of Lublin, 20079 Lublin, Poland; przemyslawsikora@umlub.pl

**Keywords:** cystinosis, corneal deposits, cysteamine, nephropathic cystinosis, juvenile cystinosis

## Abstract

Background: We aimed to identify diagnosed cases of ocular cystinosis and describe clinical, epidemiological and therapeutic characteristics. Methods: This is a descriptive and retrospective case series. All patients underwent a full check-up examination every 4–6 months by ophthalmologists, nephrologists and other required specialists. Results: Of the seven cases, six (85.7%) were females and one (14.2%) was male. The infantile nephropathic form of cystinosis was observed in five patients and the juvenile nephropathic form in two patients. No patients with the ocular form of cystinosis were identified. Corneal cystine crystals (CCC) were found in all analyzed patients. Severe ocular and general complications of the disease that had been standing for years, connected to the infantile nephropathic form, delayed diagnosis or inappropriate treatment, were observed only in two patients. All patients received topical therapy. No adverse events related to the therapy were observed. Conclusions: Cystinosis is a rare, progressive disease. Early diagnosis and treatment prevent serious complications from numerous systemic organs. Patients require constant systematic monitoring by various specialists.

## 1. Introduction

Cystinosis is an autosomal recessive inherited lysosomal storage disease that occurs in approximately 1–2 of 100,000 live births [1]. It is a multisystem disease leading to nephropathic, ocular, endocrinologic, gastrointestinal, muscular and neurological symptoms.

The first description of cystinosis was published in 1903 by the Swiss biochemist and physiologist Emil Abderhalden [2]. He described cystine crystals in the liver and spleen of a 21-months-old child, who died from dehydration and failure to thrive [3]. In 1998, a mutation in the *CTNS* gene that maps to chromosome 17p13 was firstly associated with this rare disease [2]. Since then more than 100 mutations of the *CTNS* gene have been reported, including deletions; small insertions; duplications; missense, nonsense and splice site mutations; mutations in the promotor sequence; and genomic rearrangements [1,2,3]. *CTNS* gene encodes the lysosomal membrane transporter protein which is responsible for cysteine transport in cells [3,4]. Mutations in the *CTNS* gene result in defective transport of cysteine out of the lysosomes into the cytoplasm. It leads to its excessive accumulation within cellular lysosomes, and subsequent intracellular crystal formation due to its high insolubility [4]. As the disorder progresses, cystine crystal deposits accumulate in multiple organs, initially in the kidneys and eyes later in thyroid, testes, pancreas, muscles and brain [5].

Currently, different diagnostic tools are available for diagnosis of cystinosis. The first one is the identification of characteristic accumulated corneal cystine deposits on slit lamp biomicroscopy by an experienced ophthalmologist. This diagnosis is supported by testing of the free non-protein cystine level in white blood cells of peripheral blood [6,7]. The normal cystine level in leukocytes is <0.2 nmol half-cysteine/mg protein. The third method is genetic analyses which is very helpful, but it is not essential for diagnosis.

Though it is a monogenic disease, we distinguish three different types of cystinosis, which differ in clinical presentation and severity of mutation. The most frequent phenotype is the infantile nephropathic form. The juvenile nephropathic form is less common and the ocular non-nephropathic one extremely rare. [8]. Ocular manifestations are observed in all three forms of cystinosis.

Corneal crystal deposits are the main ocular findings. They can appear as early as 2 months after birth, but usually do after 12–16 months [5,7], initially in the corneal periphery and later centrally and posteriorly [9].

Cystine crystals accumulate in all ocular tissues leading from mild to severe complications. Superficial punctate keratopathy, recurrent erosions, filamentary keratitis, band-shaped keratopathy, peripheral neovascularization, superficial limbal calcification, limbus stem cell deficiency and ulcers can be observed [8]. Severe complications can sometimes lead to keratoplasty. A less frequent complication is glaucoma due to cystine crystal storage in the trabecular meshwork or angle closure with pupillary block due to posterior synechiae [10]. Hyper- or hypopigmentary retinopathy, photoreceptor degeneration and papilloedema sometimes manifest in funduscopic examination [11,12].

Initially patients do not report any ocular symptoms, and as the disease progresses and corneal sensitivity is changed, they start to complain of photophobia, glare disability and decreased contrast sensitivity. The most irritating symptoms are stinging, epiphora, blepharospasm and vision impairment due to a large number of corneal deposits.

Disease progression is slowed by topical and oral cysteamine treatment. Topical therapy is essential in cases with ocular involvement. Cysteamine eye drops dissolve the crystals and alleviate the symptoms. To date there are two kinds of eye drops: hospital preparation eye drops with 0.5% cysteamine or ready ones (Cystadrops™, Orphan Europe, France). Cystagon™ (mercaptamin) also distributed by Orphan Europe is the only available medication used in the systemic therapy of cystinosis. However, oral administration of cysteamine does not prevent corneal complications because it has no effect on corneal cystine crystals [5].

The aims of this study are to identify diagnosed cases of ocular cystinosis; to describe the clinical, epidemiological and therapeutic characteristics in a tertiary care center in Poland; and to present a literature review, according to an evidence-based medicine approach.

## 2. Materials and Methods

It is a descriptive and retrospective case series. The study was developed at the General Department of Ophthalmology of Medical University of Lublin (Poland) from January 2010 to February 2020. The study, compliant with the tenets of the Declaration of Helsinki, was approved by the Institutional Review Board (n° KE-0254/224/2020EC). Informed consent from patients was given.

Recruiting a full cohort of affected patients currently living in Poland was challenging. Data of four not included patients, including age, gender, diagnosis based on clinical symptoms, molecular testing and cystine leukocyte counts are known from available medical history. These patients were not able to be followed-up every 4–6 months in our department, so they were not included in our study. However, it is worth emphasizing that all children aged 9–15 years with cystinosis living in Poland are described in this article. Two of them, patients P1 and P2, are sisters.

The following data were collected from information contained in the electronic medical records: epidemiological (such as age and gender), diagnostic aspects related to the disease (such as clinical pictures, blood tests and molecular test results) and response and compliance to topical treatment (hospital-formulated versus ready-to-use product eye drops).

Full ophthalmologic examinations were performed, including best-corrected visual acuity (BCVA), tonometry, anterior and posterior exams, optical coherence tomography (OCT) of anterior (AS-OCT) and posterior segments (PS-OCT) and confocal in vivo microscopy (IVCM). Additionally, photographs of anterior and posterior segments were taken every 4–6 months. A refraction test was performed every year.

Visual acuity was determined for each eye separately. BCVA was measured using early treatment diabetic retinopathy study charts by a single well-trained and experienced ophthalmologist (M.K.). Vision results were quantified as a logarithm of the minimum angle of resolution (logMAR).

All patients underwent conjunctival examination to assess the presence of cystine crystals and level of conjunctival hyperemia (Figure 1 and Figure 2). The degree of corneal crystal deposition was assessed using Gahl’s corneal cystine crystal (CCC) score on slit lamp examination and photographs on a scale 0–3, with steps of 0.25 (based on opacity, density and number of crystals) during first visit (FV) and last visit (LV).

Anterior segment optical coherence tomography (AS-OCT) images were obtained using OCT Casia Swept Source 1000 (Tomey, Inc., Nagoya, Japan). This system achieves high resolution imaging of 10 (axial) and 30 µm (transverse) and high-speed scanning of 30,000 A-scans/s. In addition, the Swept Source system minimizes the motion artifacts due to eye movement and/or faulty fixing of the eye. The depth of crystal accumulation (µm) in the central cornea was estimated using the measurement calipers provided by the AS-OCT software and it was expressed as a percentage of the corneal thickness.

In vivo confocal microscopy (IVCM) was also used to evaluate crystal deposits in each layer of the central cornea. These high-resolution images of the cornea were provided by laser scanning in vivo confocal microscopy, Heidelberg Retina Tomograph II with Rostock Corneal Module (HRT2-RCM, Heidelberg Engineering, Heidelberg, Germany). Patient examination was conducted over anesthetized cornea. Using a poly-methyl-methacrylate (PMMA) disposable cap and gel was mandatory for proper corneal applanation. Scans were collected from epithelium to endothelium. IVCM score was used to quantify the crystal deposition over a 400 × 400 µm area of the central cornea in the epithelium, Bowman layer, anterior stroma, deep stroma and endothelium. IVCM score was classified as 0 if no crystals, 1 if covering 25%, 2 = 25–50%, 3 = 50–75% and 4 = over 75% of the determined area.

Fundus examination was performed after dilation of each eye with 1% tropicamide. Fundoscopy was documented using a fundus camera and posterior segment optical coherence tomography (PS-OCT) to picture the optic disc and macula once a year.

The patients’ ocular complaints were also noted. The degree of photophobia, stinging and blepharospasm were recorded. Clinician- and self-assessed photophobia were evaluated using Liang cystinosis photophobia scaling system, previously published for, among others, vernal keratoconjunctivitis and other ocular surface inflammations [13]. Self-assessed stinging and blepharospasm were noted on the scale (none–light–mild–severe).

All ophthalmological examinations were conducted under controlled standardized room illumination conditions.

The exclusion criterion was unavailability of the medical records.

Data on each patient were evaluated; median and range of the most important parameters were calculated and transferred on to a Microsoft Excel 2010 spreadsheet, where results were arranged in a tabulated format.

## 3. Results

Fourteen eyes of seven patients (53.8%) with cystinosis, corresponding to 13 of the patients with cystinosis currently living in Poland, were evaluated and enrolled in the study. Six were females (85.7%), one was a male (14.2%). Among them, four (57.1%) were children aged between 9 and 15 and two of them were sisters at age 9 and 12. All children were females. The average age of all patients was 20.5 years (Table 1).

The infantile nephropathic form of cystinosis was observed in five (71.4%) patients, and the juvenile nephropathic form in two (28.6%). There was no patient with the ocular (non-nephropathic) form of cystinosis (Table 1).

Most of examined patients were referred from nephrologists to confirm suspicion of cystinosis due to their mild renal failure (Table 2).

In our study only one patient (P7) with the infantile form of cystinosis was diagnosed very late at 29 years of age. This may have been due to the disease being rarely recognized in the 1980s. Generally, ages of patients at diagnosis were between the 1st and 29th years of life with an average of 9.7 years (Table 1).

All other patients had siblings, but not everyone was given genetic testing. Though we have information that they were all healthy, there is a 25% risk of being a carrier (Table 1).

All patients underwent genetic analyses that revealed various mutations in the *CTNS* gene, but interestingly the presence of a 57 kb deletion was found only in two patients (P1, P2). This kind of mutation is the most characteristic and the most frequently described mutation in the *CTNS* gene that causes cystinosis (Table 1).

We noted abnormal, elevated cystine concentration in leukocytes in all examined patients. Values of 1.02–7.7 nmol half-cysteine/mg protein were recorded at the time of diagnosis. We observed significantly lower levels of leukocyte cysteine in two patients with the juvenile form, 1.02 (P4) and 2.53 (P5) respectively, compared to other patients with infantile form (Table 1).

We recorded various ocular symptoms that patients complained about. Photophobia was the most common complaint which was increased with age and amount of cystine crystal deposition. Others like stinging, blepharospasm, epiphora and foreign body sensation were also found but less often. There were no complaints of stinging and blepharospasm in patients P1, P2, P4 and P5. Patient P3 complained of light to no stinging and no blepharospasm. The most serious complaints (severe) of stinging and blepharospasm were related to patient P6. Patient P7 sometimes displayed light stinging and blepharospasm (Table 3).

Clinically assessed photophobia was mainly grade 0 (*n* = 3), 1 (*n* = 1), 2 (*n* = 1) and 3 (*n* = 1). Blepharospasm grade was none (*n* = 5), light (*n* = 1) and severe (*n* = 1). The significant degrees of photophobia and blepharospasm in two patients (P6, P7) were correlated with severe ocular complications.

In three patients (P1, P2, P3) the BCVA was normal and stable during all follow-up period. In one patient (P4), amblyopia due to refractive error and subsequent esotropia were addressed by being operated on in childhood, and that led to a low BCVA of 0.9 logMAR in the right eye (RE). A low level of myopia of about −0.5 D in left eye (LE) was found in one patient (P5). The most serious impairment of visual acuity was observed in one patient (P6) and was due to anterior and posterior segment complications (Figure 3). His BCVA progressively deteriorated to −1.8 logMAR and −2.8 logMAR (LV) in RE and LF, respectively, at LV. The first keratoplasty in RE was performed in 2018 due to bullous keratopathy as a result of corneal decompensation and the second one in 2020 because of the transplant rejection. During the second surgery, pars plana vitrectomy on account of retinal detachment and cataract extraction were performed in RE. Retinal detachment was a result of a RE injury. Complaints about visual impairment were observed as well in one patient (P7). Accumulation of large amounts of cystine crystals in the cornea and subsequent decompensation were the cause of keratoplasty in LE in 2013 (Table 2). His BCVA was 0.7 in his RE and 0.3 in his LE, and he was stable for FV and LV.

Corneal and conjunctival deposits were found in all patients enrolled. Inpatient P7 cystine deposits were visible in the iris as single, reflective crystals. The locations of visible cystine crystals varied across the cornea; however, they were identified in the center and periphery, and the anterior and deep stroma of the cornea.

Results of CCC score assessed in FV and LV with a biomicroscope and slit lamp photographs are displayed in Table 3. We did not detect cystine crystals in patient P1 on her FV at the age of nine months (CCC score = 0.00). They started to appear at around 2 years old (CCC score = 0.25–0.5). We observed an increase in accumulation of crystal deposits with age despite the 0.5% cysteamine eye drops (CCC score = 0.75–2.00). A similar increasing number of deposits occurring with age was also recorded in her older sister (patient P2). The worsening of CCC score could be connected with a very poor compliance. In fact, despite thorough care of parents, it was hard for them to sustain a daily frequent dose regimen of 8–10 times a day. Estimation of corneal deposits using CCC score and AS-OCT was very difficult in patient P6 due to long-term anterior segment complications which changed the picture of cornea layers.

First successful AS-OCT exam was conducted in patient P1 when she was 4 years old. Our patients manifested with hyperreflective deposits occurring at different depths of the cornea. On the first visit crystals were detected just in the anterior stroma in three patients (P1, P4, P5). In the remaining ones (P2, P3, P6, P7), cystine deposits were also located in deep stroma, in places covering the entire cornea thickness (Figure 4). In one patient (P6) it was difficult to evaluate accumulated crystals over the years with AS-OCT, due to morphological changes of cornea and its increased thickness. However, they were hardly visible.

In one patient (P6) it was difficult to evaluate accumulated crystals over the years with AS-OCT, due to morphological changes of cornea and its increased thickness. However, they were hardly visible. On the last visit, depositions of crystals were slightly less in three patients (P4, P5, P7). Significant decreases in amount of crystals we recorded in three patients (P1, P2, P3). This difference resulted from different ocular therapy implemented.

Values of cornea thickness (µm) were very similar on the FV and LV: 520–735 and 526–944, respectively. The greatest values (735 and 944 µm) were noted in one patient (P6), but in the remaining patients’ values of cornea thickness were similar. The depth of crystal accumulation (in µm) in the central cornea was expressed as a percentage of the corneal thickness (Table 4).

Assessment of crystal density using IVCM was a more precise method; however, it was very difficult to perform on the youngest patients (P1, P2, P3). We recorded needle-shaped, spindle-shaped and fusiform hyperreflective crystals in different cornea layers (Figure 5).

The IVCM scores obtained from the epithelium, Bowman layer, anterior stroma, deep stroma and endothelium in all eyes ranged between 3 and 12 (Table 4).

We noticed absence of retinal cystine crystals and no other abnormalities in the retina in five patients (P1, P2, P3, P4, P5). Only in one patient (P7) did we observe small hyperpigmentation in macula of LE. Funduscopic examination was difficult in the observation of one patient (P6) due to not very translucent cornea after penetrating keratoplasty, a large number of corneal deposits in the second eye and band keratopathy which obscured the picture of the fundus.

We did not observe symptoms of glaucoma in any of the patients. Tonometry, optical coherence tomography (OCT) of optic disc and photographs of optic disc were within normal limits in all patients.

We observed ophthalmologic and nephrologic symptoms in all patients. Endocrinologist’s consultation was required in one of the youngest patients (P3, aged 13) due to the growth retardation. Many complications of cystinosis related to various organs were diagnosed in patients P6 and P7. They required diabetologic (P6, P7), endocrinologic (P6, P7), neurologic (P6), hepatologic (P7), orthopedic (P7) and gastroenterologic (P7) treatment (Table 2).

Five patients (P1, P2, P3, P6, P7) with the infantile form of cystinosis were treated with orally administered cysteamine (Cystagon™) 4 times a day. Two patients (P3, P6) were not given Cystagon™ constantly. In two patients (P4, P5) with the juvenile form of cystinosis, Cystagon™ was not used. Three patients (P1, P2, P3) were initially treated by hospital formulated 0.5% cysteamine eye drops applied 2–10 times a day and later four times a day. Other patients (P4, P5, P7) applied a hospital formulated preparation of 0.5% cysteamine eye drops 2–10 times a day during the entire period of observation. One patient (P6) applied cysteamine eye drops just during 2016 and discontinued because of the strong burning after use and severe ocular complications (Table 2). We noticed a significant decrease in corneal cystine crystals (CCC score, AS-OCT, IVCM) after administration of 0.55% Cystadrops compared to hospital formulated preparation of 0.5% cysteamine eye drops in three patients (P1, P2 and P3; Figure 6).

## 4. Discussion

Cystinosis is categorized as a rare disease [1,14]. with a prevalence of 1–2 persons in 100,000 live births. The rate of cystinosis in Poland is 1 person in 2.92 million in the Polish population. Based on literature data, the highest rate of cystinosis in the world is found in French Canada (1:62,500 live births) [8,15]. Incidence rates reported in France, Germany, Denmark, Sweden and Australia are 1:167,000, 1:179,000, 1:115,000, 1:260,000 and 1:192,000 live births, respectively [8]. In Europe, higher incidence is observed in Brittany in France (1:26,000 live births) [16]. The only known country where cystinosis has never been confirmed is Finland [16]. Cystinosis is found worldwide in all ethnic groups but its geographical distribution is unequal [7]. Based on current data of the Polish Registry of Hereditary Tubulopathies (POLtube), only 13 diagnosed patients live in our country but the exact incidence of cystinosis in the Polish population is still unknown. This may be connected with the small detection rate of this disease.

Cystinosis is a lysosomal storage disorder that contributes to the disruption of cystine transport out of the lysosomes [17]. To date, mechanisms of tissue damage are not fully understood. Studies have shown that cells are damaged due to altered mitochondrial metabolism, autophagy and apoptosis [18].

Due to the rarity of the disorder, diagnosis is often delayed, which results in severe prognosis of the condition [4]. The infantile form of cystinosis is usually diagnosed during the first 2 years of life [4]. In our case series, diagnosis of the infantile nephropathic form was significantly delayed only in one patient (P7); he was 29. Others ranged between 1 and 2 years old. Detection of the juvenile nephropathic type usually happens in the second decade of life like in our two patients (P4, P5) [3,8]. Infantile nephropathic cystinosis is the most common form accounting for approximately 95% [1,6,8]. This is the most severe form, manifested with Fanconi syndrome (proximal renal tubulopathy), failure to thrive, rickets and progression to end stage kidney disease (ESKD) in the first decade of life if not treated [3,8]. Infants with nephropathic cystinosis usually have pale, blond hair, fair eyelashes, blue eyes and fair skin due to pigmentation defects [1]. The patients usually show growth retardation, rickets, myopathy, hypogonadism, diabetes mellitus, swallowing difficulties, pulmonary disfunction and central nervous system damage [8,11]. The juvenile nephropathic cystinosis form is usually diagnosed later in childhood or adolescence and accounts for less than 5% of cases. It is a benign form of cystinosis limited to mild renal disease and corneal manifestations [3]. The third form of cystinosis is ocular (non-nephropathic) and it is diagnosed in adult patients. Its occurrence is extremely rare [17]. It is characterized by corneal cystine crystal depositions without renal anomalies [3].

In our study, the infantile nephropathic form accounted for 71.4% patients and the juvenile nephropathic form about 28.6%. There was no case with the ocular form of cystinosis, and clinical symptoms in patients with infantile cystinosis were more severe than in patients with juvenile form.

The current gold standard in diagnosis of cystinosis is the detection of elevated cystine concentration in white blood cells, which is a very sensitive and specific method [4]. Oshima et al., as a first, established measurement of cystine in leukocytes in 1974 [8]. Ariceta et al. reported values of >1 nmol ½ cystine/mg protein (usually > 2) and Broyer et al. values of 3–23 nmol ½ cystine/mg protein in affected individuals with nephropathic cystinosis at the time of diagnosis [7,19]. Similar leukocyte cystine concentrations ranging from 3.2 to 7.7 nmol ½ cystine/mg protein in patients with infantile cystinosis were detected in our study. Noteworthy is the fact that lower values of 1.02–2.53 nmol ½ cystine/mg protein were observed in our patients with juvenile cystinosis. That may have been due to the much milder course of the juvenile form compared to infantile cystinosis. Currently, additional several biochemical methods are used for cystine concentration measurement. Few studies have detected cystine concentration values in granulocytes, as cystine preferentially accumulates in these cells [11,20].

It is thought that as much as 80% of rare diseases are genetic. Cystinosis is inherited as an autosomal recessive trait in all variants.

Each form of cystinosis shows different mutations in the *CTNS* gene [21]. Molecular genetic testing in diagnosis is very efficient, but in 5% of patients, mutations in *CTNS* gene are not easily discovered [8]. The most frequent mutation accounting for approximately 75% of the affected patients in the European population is a large 57 kb deletion, affecting the first 10 exons of *CTNS* [3,21]. Our molecular diagnosis revealed the same mutation in two sisters (P1, P2). Al.-Haggar et al. reported a correlation between genotype/phenotype and the clinical types of cystinosis. Severe mutations were found mostly in patients with infantile cystinosis compared to patients with juvenile and ocular types [21].

Prenatal diagnosis is available in carriers and it allows for early detecting of the disorder and early cysteamine therapy. Cystinosis, affecting many organs, leads to multisystemic dysfunction. Though this article mainly focuses on ophthalmologic aspects related to patients affected by the ocular manifestations of cystinosis, systemic clinical history was recorded too. The most severe systemic symptoms were exhibited by the oldest patients (P6, P7) with the infantile form of cystinosis. ESKD resulted in renal transplantation at age 7 and 30 in patient P7, and in patient P6 when she was nine years old due to very delayed diagnosis and late implemented cysteamine treatment. Data from the scientific literature confirm the development of ESKD by 10 years of age if the disease is untreated [1,7]. In our series, renal manifestations in patients with the juvenile form were mild and limited to isolated tubular proteinuria (P4, P5) and glycosuria with partial Fanconi syndrome (P4). However, Al.-Haggar et al. reported a wide spectrum of renal symptoms, from mild proximal tubulopathy to ESKD but with a slower rate of progression in the juvenile form compared to infantile cystinosis [21].

Ocular involvement is the first extrarenal symptom of cystinosis [8]. Depositions of cystine crystals in the cornea and conjunctiva are pathognomonic symptoms of the condition [9]. Data from the literature report accumulation of cystine deposits in the iris, ciliary body, choroid, retina and optic nerve too [22]. Only one (P7) out of the seven studied patients presented cystine crystals not only in the cornea and conjunctiva but also in iris. Our findings showed crystal deposition within not only the anterior stroma but also deeper, covering the entire cornea depth-wise. Locations in anterior stroma were observed in patients with juvenile cystinosis and in the youngest patients. Labbe et al. reported a mean depth of 291.4 µm [23]. A previous study has reported the appearance of cystine crystals mainly within the anterior stroma [23]. Our study demonstrated that AS-OCT is a non-invasive method that can be used easily, even in very young patients, and it is very useful to visualize cystine crystals. However, any ocular surface complications occurring, such as scars, band keratopathy, bullous keratopathy, etc., are a limitation in the evaluation of crystal location.

IVCM is the gold standard for ophthalmologic follow-up in cystinosis [24] and the only imaging technique able to demonstrate CCC in vivo [13,25]. However, being a contact technique, it is very difficult to perform in the youngest patients. Additionally, to obtain good images, the corneal surface has to remain in good contact with the objective lens during the whole examination. A previous study has shown that there is no significant difference between measurements of cystine crystals depth and central cornea thickness obtained with IVCM and AS-OCT [23]. IVCM findings from the literature report nerve abnormalities too, such as enlarged and abnormal-looking tortuous nerves and beaded nerves in the sub-basal plexus of the cornea in cystinosis patients [26]. In our study, IVCM showed a greater amount of CCC that increased mainly with age.

Ocular symptoms were less severe in children with infantile cystinosis (P1, P2, P3) and with the juvenile form compared to two the oldest patients (P6, P7) with more severe systemic symptoms as well. In addition to corneal involvement, hypo- and hyperpigmentation of retina may be identified. In 10–15% patients, retinopathy leads to blindness [8]. In our study group, just one patient (P7) manifested with small hyperpigmentation in macula.

Deterioration of visual acuity is another finding related to cystine crystals in the cornea observed in some patients. We have not performed photopic and scotopic ERG to check the functionality of rods and cones in studied patients. Ariceta et al. reported that there is no need to test ERG in patients who do not demonstrate altered night vision and whose retinal examination is within normal limits [11].

To date, cysteamine therapy in the form of cysteamine bitartrate (Cystagon™) is the only and essential treatment which revolutionized the management and prognosis of patients with nephropathic cystinosis. Cysteamine was approved for clinical use in the USA and Europe in the 1990s [6]. It is cystine depletion therapy because it can deprive cystinotic cells of more than 90% of the accumulated cystine [7]. As 100% of patients with each form of cystinosis manifest corneal accumulation of cystine deposits, topical cysteamine therapy is also recommended. Indeed, oral administration of cysteamine has no effect on accumulated corneal crystals because of poor systemic delivery to the avascular cornea, but it may reach the retinal epithelium, thereby preventing retinal infiltration [17,26]. It seems that cysteamine eye drops dissolve cystine crystals, preventing anterior eye segment complications [2], which we observed in two of our patients (P6, P7) due to late implemented and discontinued treatment. The two youngest patients (P1, P2) with infantile cystinosis who were diagnosed early and treated according to recommendations, showed normal psychomotor and physical development and renal function. In contrast, a 12-year-old female patient (P3) with poor medication compliance, despite the early diagnosis, manifested with growth retardation and significant corneal crystal deposition. In fact, infantile cystinosis patients grow at a normal rate if they are treated properly, as reported in the literature [7].

Currently, topical therapy comprises two kinds of eye drops—pharmacy formulated 0.5% cysteamine hydrochloride and licensed preparation of 0.55% Cystadrops™ (Orphan Europe, Puteaux, France) distributed in Europe and 0.44% Cystaran™ (Hi-Tech Pharmacal Co., Amityville, NY, USA) in the USA. Cystadrops™ has been approved by the European Medicines Agency (EMA) for use in the European Union in 2017 for patients with ocular cystinosis aged 2 years or more [27]. This specific topical treatment should be initiated under the supervision of an ophthalmologist experienced in the management of cystinosis. Off-license 0.5% hospital preparation eye drops as an aqueous solution with no viscosity agent should be administered very frequently 6 to 12 times/day to improve corneal bioavailability. Cystadrops™, a new gel solution of cysteamine hydrochloride, thanks to their high viscosity, allow deeper penetration into the cornea; and longer corneal contact allows a less frequent dosing regimen, 4 times/day, and dissolves corneal crystals more effectively and faster [13,28]. The most frequently reported ocular adverse reactions, occurring in >10% of patients taking Cystadrops™, were eye irritation, blurred vision, ocular hyperemia, eye pruritus and increased lacrimation. We observed faster dissolving of conjunctival and CCC and no ocular adverse reactions with administration of Cystadrops™ compared to pharmacy preparation eye drops in three patients (P1, P2, P3). Our observations are in accordance with reports in literature [4,13]. Unfortunately, topical cysteamine therapy has no effect on existing band keratopathy, corneal pannus or scars because crystals rarely contribute to the formation of those [2]. It has been shown that, if the treatment is discontinued, accumulation of CCC increases [6].

Cysteamine treatment is very expensive due to rarity of cystinosis and relatively small group of affected patients. Access to oral and topical cysteamine therapy has been improved only recently in Poland and in other European countries. Currently, in Poland patients with the nephropathic form of cystinosis have the opportunity to receive Cystagon™ and Cystadrops™ under the Emergency Drug Access Program. Decision to start this treatment is based on the patient’s current clinical symptoms and is made by the Ministry of Health individually for each of patients, and it requires renewal every 3 months. This limited access to cysteamine therapy may have had a considerable impact on the course of illness and prognosis in recent decades. Moreover, the transition of patients from pediatric to adult care providers requires special attention and is a challenge in correct management of adult patients with cystinosis. The rare nature of cystinosis limits the capacity for large clinical trials.

In conclusion, since the introduction of its treatment, long-term prognosis of cystinosis has been improved. Close cooperation between specialists of different areas of medicine is essential in efficient management of patients with cystinosis, including diagnosis, follow-up and appropriate treatment.

## Figures and Tables

**Figure 1 diagnostics-10-00911-f001:**
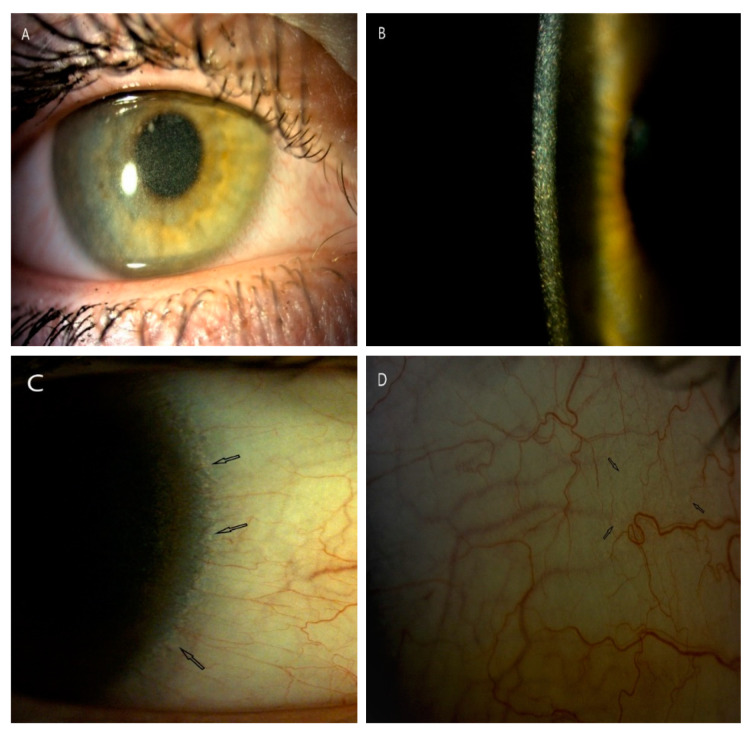
Corneal slit-lamp photographs (10–25×) of a patient (P5) with juvenile form of cystinosis. (**A**) View of whole cornea with crystal deposits in wide slit-lamp beam. (**B**) A large number of cystine deposits extending from the center to the periphery of the cornea in high magnification slit beam (25×). (**C**) Arrows show cystine deposits in corneal limbus and in conjunctiva (**D**).

**Figure 2 diagnostics-10-00911-f002:**
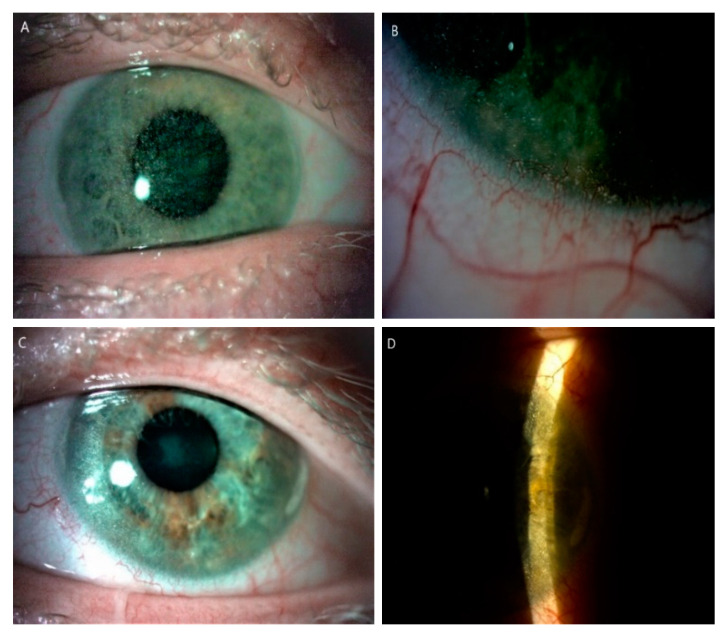
Slit-lamp images (10–25×) of posttransplant cornea in a patient (P7) with an infantile form of cystinosis. (**A**) Diffused cystine crystalsin right eye. (**B**) Peripheral corneal neovascularization with crystal deposits in the left eye after keratoplasty. (**C**) Cystine crystals are visible in the host cornea, in wide slit beam and (**D**) narrow slit beam in left eye.

**Figure 3 diagnostics-10-00911-f003:**
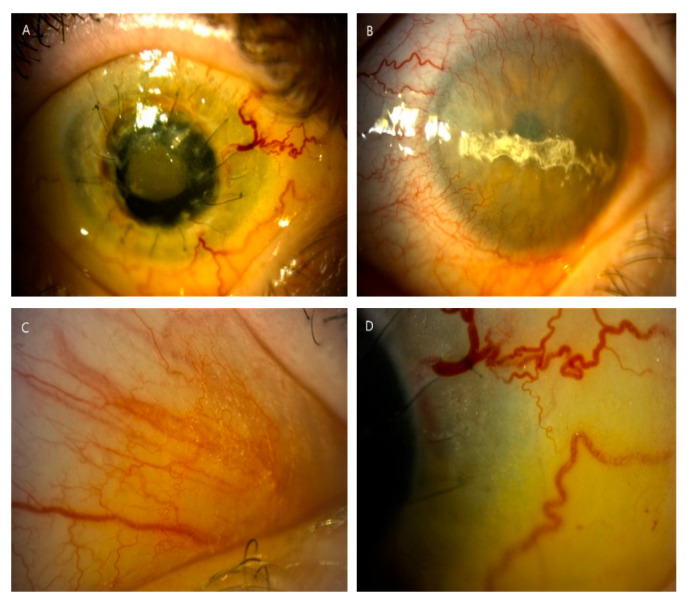
Slit-lamp photographs (10–25×) displayed complications of the anterior segment in long-term nephropathic cystinosis with delayed treatment (patient 6). (**A**) Cornea after two unsuccessful keratoplasties with hazy transplant, ingrown vessels, loose sutures and hyperemia of conjunctiva in the right eye. (**B**) Band keratopathy and significant peripheral corneal neovascularization in the left eye. (**C**,**D**) Significant density of crystal deposits in conjunctiva, in places blurring the drawing of the vessels.

**Figure 4 diagnostics-10-00911-f004:**
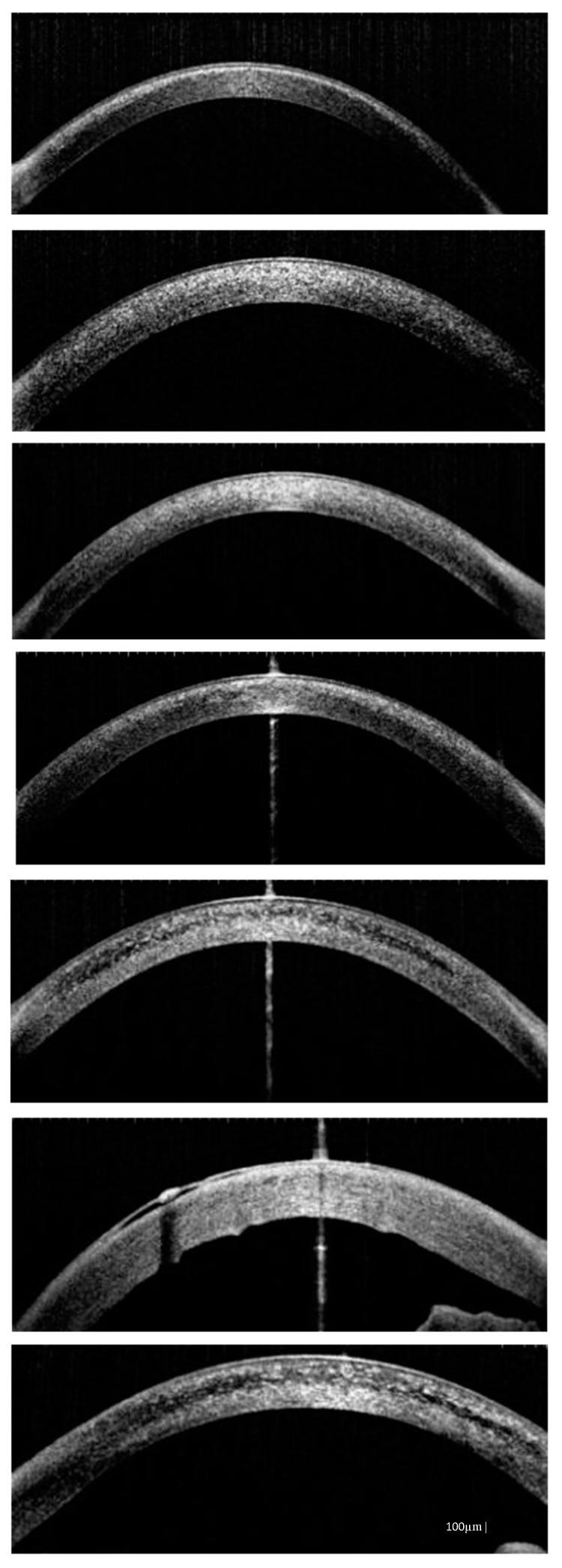
Anterior segment optical coherence tomography (AS-OCT) images showing deep corneal cystine crystals (CCC) in studied patients (from the top: P1, P2, P3, P4, P5, P6, P7). Band keratopathy and thickened cornea are visible in patient P6 as complications of long-term cystinosis.

**Figure 5 diagnostics-10-00911-f005:**
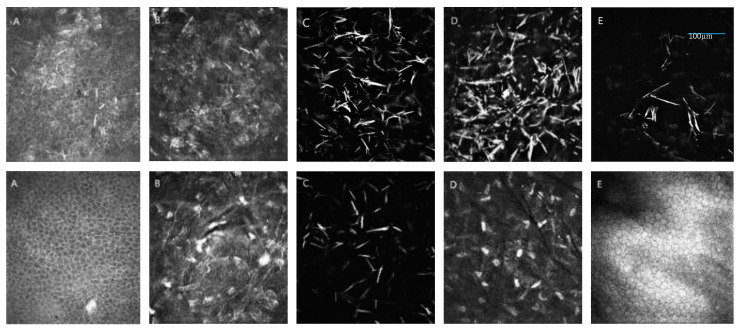
Presentation of cystine crystals in in vivo confocal microscope (IVCM) images over a 400 × 400 µm area. On the top, significantly greater accumulation of cystine deposits in patient P3 with infantile cystinosis compared to (below) patient P4 with juvenile cystinosis. (**A**) Epithelium, (**B**) Bowman layer, (**C**) anterior stroma, (**D**) deep stroma, (**E**) endothelium. Additionally, panel A shows intracellular (IC) and extracellular (EC) crystals in the epithelium layer.

**Figure 6 diagnostics-10-00911-f006:**
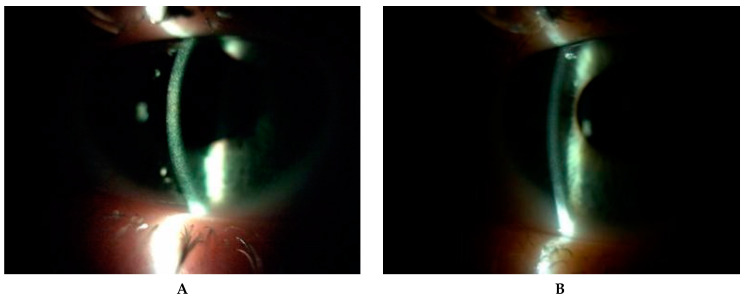
Corneal cystine crystals in slit lamp (10×) before administration of hospital formulated 0.5% cysteamine (**A**) and one year after application of 0.55% Cystadrops (**B**) in patient 2.

**Table 1 diagnostics-10-00911-t001:** Cystinosis presentation divided into two age groups. Genetic analyses confirmed diagnosis of cystinosis in all patients. Cysteine level in leukocytes at the time of diagnosis was above normal content in all patients. Infantile nephropathic form was in 5 patients and juvenile form in 2.

Patients	Group Age	Year of Birth	Time of Diagnosis (Year)	Genetic Testing *CTNS* Gene Mutations	Cystine Level (nmol ½ cysteine/mg Protein) normal <0.2 at the Time of Diagnosis	Forms of Cystinosis			Siblings
						Infantile Nephropathic	Juvenile Nephropathic	Ocular	
**P1**	9–15 years	2011	2012	hom 57kb del	7.7	+			1 affected sister P2
**P2**		2008	2010	hom 57kb del	4.6	+			1 affected sister P1
**P3**		2007	2009	+	3.2	+			1 stepbrother
**P4**		2005	2017	+	1.02		+		2 healthy sisters; no gen. test
**P5**	24–37 years	1996	2016	+	2.53		+		1 healthy sister; no gen. test
**P6**		1986	1988	+	4.1	+			1 healthy sister; gen test made
**P7**		1983	2012	+	5.5	+			3 healthy sisters gen. test made

hom: homozygosity; Kb: kilobyte; del: deletion; nmole: nanomole; +: positive finding.

**Table 2 diagnostics-10-00911-t002:** Results of ophthalmological exams with general symptoms and treatment.

Patients	BCVA (logMAR)		Tonometry (mmHg)	Consultations	General Symptoms	Cystinosis Local Treatment	Cystinosis Oral Treatment
	FV	LV	LV				
**P1**	RE 0.0 LE 0.0	RE 0.0 LE 0.0	RE 13 LE 16	Ophthalmological Nephrological	Fanconi syndrome	0.5% Cysteamine 2011 0.55% Cystadrops since 2018	Cystagon 2011
**P2**	RE 0.0 LE 0.0	RE 0.0 LE 0.0	RE 15 LE 14	Ophthalmologica Nephrological	Fanconi syndrome	0.5% Cysteamine 2011 0.55% Cystadrops since 2018	Cystagon 2010
**P3**	RE 0.0 LE 0.0	RE 0.0 RE 0.0	RE 14 LE 17	Ophthalmological Nephrological Endocrinological	Fanconi syndrome, growth retardation	0.5% Cysteamine 2011 (not constantly) 0.55% Cystadrops since 2019	Cystagon 2011 (not constantly)
**P4**	RE 0.9 LE 0.0	RE 0.9 LE 0.0	RE 12 LE 15	Ophthalmological Nephrological	Tubular proteinuria and glucosuria, partial Fanconi syndrome	0.5% Cysteamine since 2017	-
**P5**	RE 0.0 LE 0.0	RE 0.0 LE 0.0	RE 15 LE 11	Ophthalmological Nephrological	Isolated tubular proteinuria	0.5% Cysteamine since 2017	-
**P6**	RE -1.8 LE 1.0	RE -2.8 LE 1.0	RE 15 LE 20	Ophthalmological Nephrological Diabetological Endocrinological Neurological	Renal transplantation, diabetes, hypothyreosis, hands myopathy, growth retardation, bronchial asthma, secondary anemia, cerebral circulation failure, ectopic pregnancy, dialysis	0.5% Cysteamine during one year 2016	Cystagon 1989-1994 now since 2016
**P7**	RE 0.16 LE 0.52	RE 0.16 LE 0.52	RE 10 LE 11	Ophthalmological Nephrological Diabetological Endocrinological Hepatological Orthopedic	Renal transplantation, diabetes, hypothyreosis, growth retardation, osteoporosis, femoral head necrosis, R knee and spine arthrosis, compression fracture Th12, L1	0.5% Cysteamine since 2013	Cystagon 2014

BCVA: best-corrected visual acuity; logMAR: logarithm of the minimum angle of resolution; mmHg: millimeters of mercury; FV: first visit; LV: last visit; RE: right eye; LE: left eye.

**Table 3 diagnostics-10-00911-t003:** Self- and clinically-assessed photophobia grading based on the Liang photophobia scaling system and ocular complaints reported by examined patients. Assessment of corneal cystine crystals (CCC) using Gahl’s score (CCCS) and their locations assessed with anterior segment OCT (AS-OCT). Ocular complications (anterior and posterior segment) observed.

Patients	Liang Photophobia Score		Blepharospasm	Stinging		Gahl’s Score		CCC Localization		Ocular Complications
	Self-Assessed	Clinically Assessed								
			LV	FV	LV	FV	LV	FV	LV	
	FV/LV	FV/LV								
**P1**	0/0	0/0	none	none	none	0.00	1.5	AS	AS	Corneal and conjuntival cystine deposits
**P2**	1/0	1/0	none	none	none	2.00	1.5	DS	AS	Corneal and conjuntival cystine deposits
**P3**	3/1	2/1	none	light	none	2.75	2.5	DS	AS	Corneal and conjuntival cystine deposits
**P4**	0/0	0/0	none	none	none	2.00	1.75	AS	AS	Corneal and conjuntival cystine deposits
**P5**	0/0	0/0	none	none	none	2.25	2.00	AS	AS	Corneal and conjuntival cystine deposits
**P6**	3/3	3/3	severe	severe	severe	2.5	2.5	DS	DS	RE: keratoplasty (bullous keratopathy) + transplant rejection + secondary keratoplasty + cataract extraction (aphakic status) + vitrectomy (retinal detachment) LE: peripheral neovascularization, band keratopathy and limbal calcification
**P7**	3/1	2/2	light	light	light	3.00	2.75	DS	DS	RE: Cystine crystal orneal deposits in cornea and conjunctiva, iris LE: keratoplasty(CCC in host corneawith limbal neovascularization)

FV: first visit, LV: last visit, RE: right eye, LE: left eye; AS: anterior stroma; DS: deep stroma.

**Table 4 diagnostics-10-00911-t004:** Assessment of cystine crystals using anterior segment optical coherence tomography (AS-OCT) and in vivo confocal microscopy (IVCM).

Patients	AS-OCT		IVCM Crystals Density
	CCT (µm)	Depth of Corneal Crystals (µm) [%]	
**P1**	542	122 [22.5]	4
**P2**	544	296 [54.4]	8
**P3**	552	368 [66.6]	8
**P4**	520	210 [40.3]	3
**P5**	569	250 [43.9]	6
**P6**	789	Difficult to evaluate	12
**P7**	576	546 [94.7]	10

CCT: central cornea thickness; µm: microns.

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
