# Peer review of "Ophthalmic Evaluation of Diagnosed Cases of Eye Cystinosis: A Tertiary Care Center’s Experience"

_diagnostics, 2020, doi:10.3390/diagnostics10110911_

Round 1

Reviewer 1 Report

The authors must be congratulated for this well-written paper.

I only have minor changes to suggest:

  • consider including "ophthalmic evaluation of diagnosed....." in the title.
  • Abstract; lines 22-23; please change "in the study group" for "analyzed". If there is no control group (and there is not), I think it is better not to use the term "study group"

Author Response

OPEN REVIEW 1

English language and style

( ) Extensive editing of English language and style required
( ) Moderate English changes required
(x) English language and style are fine/minor spell check required
( ) I don't feel qualified to judge about the English language and style

Authors’ replay:

We thank the reviewer for the suggestions. Minor spell check has been done now for the whole manuacript as requested.

Yes

Can be improved

Must be improved

Not applicable

Does the introduction provide sufficient background and include all relevant references?

(x)

( )

( )

( )

Is the research design appropriate?

(x)

( )

( )

( )

Are the methods adequately described?

(x)

( )

( )

( )

Are the results clearly presented?

(x)

( )

( )

( )

Are the conclusions supported by the results?

(x)

( )

( )

( )

Authors’ replay:

Thanks for the positive comments.

COMMENTS AND SUGGESTIONS FOR AUTHORS

The authors must be congratulated for this well-written paper.

I only have minor changes to suggest:

  • consider including "ophthalmic evaluation of diagnosed" in the title.
  • Abstract; lines 22-23; please change "in the study group" for "analyzed". If there is no control group (and there is not), I think it is better not to use the term "study group"

Authors’ replay:

Thanks for the positive comments and valuable comments. As suggested, we have changed the title. Additionally, we have changed "in the study group" for "analyzed”. A minor spell check has been done now for the whole manuacript as requested. The flow of Introduction, Results, and Discussion sections has been improved. Moreover, all figure and tables legends have been reedited.

Reviewer 2 Report

I congratulate the authors on their work. The manuscript overall is well written. However, the readers will benefit from a more concise manuscript. The introduction and discussion sections are lengthy.

Figure 2 – please check the English – “A great number aggregated cystine crystals, creating conglomerates in right eye” and “donor one is spare”

Table 2 – please mention what does BCVA RE 1.0. Is it LOGMAR ?  

Table 3 – please expand the abbreviations such as RV, LV, RE, LE, OCT etc.

Author Response

OPEN REVIEW 2

English language and style

( ) Extensive editing of English language and style required
( ) Moderate English changes required
(x) English language and style are fine/minor spell check required
( ) I don't feel qualified to judge about the English language and style

Authors’ replay:

We thank the reviewer for the suggestions. Minor spell check has been done now for the whole manuacript as requested.

Yes

Can be improved

Must be improved

Not applicable

Does the introduction provide sufficient background and include all relevant references?

( )

(x)

( )

( )

Is the research design appropriate?

( )

(x)

( )

( )

Are the methods adequately described?

( )

(x)

( )

( )

Are the results clearly presented?

( )

(x)

( )

( )

Are the conclusions supported by the results?

( )

(x)

( )

( )

Authors’ replay:

We thank the reviewer for the valuable suggestions. As suggested, we agreed with all suggestions below and we did our best to improve the manuscript.

COMMENTS AND SUGGESTIONS FOR AUTHORS

I congratulate the authors on their work. The manuscript overall is well written. However, the readers will benefit from a more concise manuscript. The introduction and discussion sections are lengthy.

Authors’ replay:

Thanks for the valuable comments. As suggested, the Introduction, Results and Discussion sections have been summarized. Their flow has been implemented as well.

Figure 2 – please check the English – “A great number aggregated cystine crystals, creating conglomerates in right eye” and “donor one is spare”

Authors’ replay:

As suggested, we have checked the English of all manuscript and we have rephrased the legend of the Figure 2.

we have changed the title. Additionally, we have changed "in the study group" for "analyzed”. A minor spell check has been done now for the whole manuacript as requested.

Table 2 – please mention what does BCVA RE 1.0. Is it LOGMAR?  

Authors’ replay:

Thanks for the comment. BCVA was measured using Early Treatment Diabetic Retinopathy Study charts by a single well-trained and experienced ophthalmologist. Vision results were quantified as a logarithm of the minimum angle of resolution (logMAR). In Methods section we have clarify how we checked BCVA. Additionally, we have added the BCVA unit (logMAR) in Table 2.

Table 3 – please expand the abbreviations such as RV, LV, RE, LE, OCT etc.

Authors’ replay:

We agree with the comment. As suggested, all tables and figure legends have been reedited. Additionally, an explanation of all abbreviotios has been added to the notes where needed.
